# A Robust Hierarchical Estimation Scheme for Vehicle State Based on Maximum Correntropy Square-Root Cubature Kalman Filter

**DOI:** 10.3390/e25030453

**Published:** 2023-03-04

**Authors:** Dengliang Qi, Jingan Feng, Yongbin Li, Lei Wang, Bao Song

**Affiliations:** 1College of Mechanical and Electrical Engineering, Shihezi University, Shihezi 832003, China; 2School of Mechanical Science and Engineering, Huazhong University of Science and Technology, Wuhan 430074, China

**Keywords:** vehicle state estimation, tire force estimation, maximum correntropy square-root cubature Kalman filter, non-Gaussian noise

## Abstract

Accurate acquisition of vehicle dynamics state information is essential for vehicle active safety control systems. However, these states cannot be easily measured, and the measurement is expensive. Conventional Kalman filters perform well for vehicle state estimation in Gaussian environments but exhibit low accuracy and robustness under practical non-Gaussian noise. Vehicle model parameter ingestion, inaccurate tire force calculation, and non-Gaussian noise from on-board sensors cause great challenges to the estimation of vehicle driving states. Therefore, this paper presents a robust hierarchical estimation scheme for vehicle driving state based on the maximum correntropy square-root cubature Kalman filter (MCSCKF) using easily measurable on-board sensor information. First, the vehicle mass is dynamically updated based on the recursive least squares (FRLS) method with a forgetting factor. Then, an adaptive sliding mode observer (ASMO) is designed to estimate the longitudinal and lateral tire forces. Ultimately, the vehicle states are estimated based on the MCSCKF under non-Gaussian noise. Two typical operating situations are carried out to verify the validity of the proposed estimation scheme. The results prove that the proposed estimation scheme can estimate the vehicle’s driving state accurately compared to other common methods. And the MCSCKF algorithm has better accuracy and robustness than the traditional Kalman filters for vehicle state estimation in non-Gaussian situations.

## 1. Introduction

Recently, the rapid development of the automotive industry has been of increasing interest. Vehicle active safety control systems such as the anti-lock braking system (ABS), electronic stability program (ESP), and electronic stability control (ESC) are widely used in commercial vehicles [1,2], which are essential for vehicle handling stability. The normal operation of the active safety system is based on the accurate acquisition of vehicle dynamics, such as vehicle speed and sideslip angle. For vehicle stability control systems, vehicle speed and sideslip angle are the key control targets that determine the vehicle trajectory [3]. In addition, tires are the only component contacting the vehicle and the ground, and the tire forces generated by them can dominate the vehicle’s motion. Therefore, prior knowledge of vehicle speed, sideslip angle, and tire forces is critical for vehicle active safety systems. However, the aforementioned vehicle dynamics states cannot be directly measured in commercial vehicles, or the measurement equipment is costly [4]. Thus, the use of easily available on-board sensor signals such as steering wheel angle, acceleration, and yaw rate to estimate difficult-to-obtain vehicle driving states and tire forces has been widely investigated by researchers [5].

Different methods for obtaining tire forces have been extensively studied by scholars [6]. Direct measurement of tire force requires special and expensive sensors, and its application in commercial vehicles is limited due to cost factors [7]. Some physical or empirical tire models are used to calculate tire forces in vehicle dynamics, e.g., the Dugoff tire model, the magic formula tire model, the brush tire model, etc. [8,9,10]. However, the calculation of tire forces requires knowledge of parameters related to tire characteristics, such as tire longitudinal and lateral stiffness, and the acquisition of these parameters often requires extensive prior testing and calibration. Utilizing available on-board sensors for tire force estimation becomes a practical and effective method. Wilkin et al. proposed an extended Kalman filter (EKF) to estimate the lateral tire force based on a 3-degree-of-freedom vehicle model, and the effectiveness of the proposed method was demonstrated experimentally, but the estimation error is difficult to guarantee because the EKF has only first-order accuracy [11]. Doumiati et al. estimated sideslip angle and tire lateral force based on a four-wheeled vehicle model and compared two estimation methods, EKF and unscented Kalman filter (UKF), and simulation tests proved that UKF has higher estimation accuracy than EKF [12]. Jin et al. proposed an interactive multi-model UKF for the estimation of lateral tire forces and sideslip angle. Simulation tests showed that this method can provide more accurate vehicle state estimation than the single-model approach, but the algorithm is more complex and difficult to apply to real vehicles [8]. Rezaeian et al. proposed a unified estimation scheme to simultaneously obtain the vertical, longitudinal, and lateral tire forces for each vehicle wheel [13]. A high-order sliding mode observer was designed by Rath et al. to estimate tire friction for ground vehicles [14]. Moreover, a neural network approach has also been applied to the estimation of tire forces [15]. Although various tire force estimation methods have been discussed, the improvement of estimation accuracy and robustness needs to be further investigated.

Vehicle driving state estimation techniques have also been widely studied, and different state estimators have been designed in the past few years. Common vehicle state estimation methods include fuzzy observers, sliding mode observers (SMO), nonlinear observers, and the Kalman filter [16]. Among them, the Kalman filter and its improved version have a wide range of applications in vehicle state estimation. A variable-structure EKF algorithm was proposed by Li et al. This method uses the change rate of the sideslip angle to design a feedback compensation mechanism, which effectively reduces the cumulative error and improves the estimation accuracy of the sideslip angle [17]. Chen et al. combined the Adaptive Neural Fuzzy Interference System (ANFIS) with the UKF to estimate the vehicle’s lateral speed in real time considering the body state and tire constraints and used the estimation results in the model control of the vehicle [18]. Vargas-Meléndez et al. first estimated the vehicle roll angle based on easily measurable sensor information using a neural network (NN) and then introduced the estimated results into a Kalman filter to filter the noise. Also, the vehicle’s nonlinear characteristics were considered, and the estimation results proved the high accuracy of the proposed method [19]. Compared to EKF and UKF, the square-root cubature Kalman filter (SCKF) is widely adopted due to its higher estimation accuracy and better numerical stability. Wan et al. combined the interactive multi-model theory with the SCKF to design a vehicle state observer containing multiple sub-models, which reduces the complexity of the algorithm while ensuring accuracy and real-time performance [10]. Although the above study is reliable, the negative effect of the noise distribution for nonlinear systems on the experimental results is often ignored. The conventional Kalman filter presupposes that the noise satisfies a Gaussian distribution. Actually, for most nonlinear physical systems, the noise is more likely to satisfy a non-Gaussian distribution than a Gaussian distribution [20,21]. Therefore, for nonlinear vehicle state observation systems, it is more practical to consider the process and measurement noise as non-Gaussian distributions (especially heavy-tailed distributions), since large outliers often exist in the measurement data of the sensors. Particle filtering (PF) was applied in vehicle state estimation due to its ability to suppress non-Gaussian noise [22]. However, the PF algorithm has high complexity, and the particle resampling stage causes a loss of sample validity and diversity, leading to sample degradation, which increases the difficulty for practical applications. The Huber-based Kalman filter has also become an option for dealing with non-Gaussian noise. In the literature, vehicle states and parameters were estimated simultaneously using Huber-based robust UKF [23], but when Huber’s weight function error is large, the filtering accuracy of this method will be reduced due to inaccurate measurement information. Traditional Kalman filtering is based on the minimum mean square error (MMSE) criterion, which has poor filtering performance in non-Gaussian environments [24]. Recently, the maximum correntropy criterion (MCC) in information theory has been introduced into the Kalman filter to deal with problems caused by non-Gaussian noise. The robust Kalman filter based on MCC has been successfully applied in other non-Gaussian scenarios and has shown excellent performance [25,26]. In particular, the maximum correntropy square-root cubature Kalman filter (MCSCKF) not only guarantees high accuracy and numerical stability but also suppresses the interference of non-Gaussian noise [27].

The vehicle mass variation has an important effect on the accuracy of the vehicle dynamics model [3,23]. Inaccurate vehicle mass will increase the estimation error for vehicular tire forces and states, so the vehicle mass variation factor should be considered. Although the above-mentioned issues have been studied by researchers, simultaneous consideration of vehicle model parameter ingestion (mass variation), tire force estimation accuracy, and non-Gaussian noise effects have not been adequately studied. Therefore, motivated by the issues mentioned above, this paper presents a robust hierarchical estimation scheme for vehicle states based on MCSCKF under non-Gaussian noise. Firstly, recursive least squares with forgetting factor (FRLS) is used to dynamically update the vehicle mass parameters. Secondly, the adaptive sliding mode observer (ASMO) is utilized to estimate the longitudinal and lateral tire forces. Finally, the vehicle driving state in a non-Gaussian environment is accurately estimated. The main contributions of the paper are as follows.

(1)The vehicle mass is first updated in real-time, and then the longitudinal and lateral tire forces for each wheel are estimated separately based on the ASMO methodology. Based on common on-board sensors, a robust hierarchical estimation scheme is designed for vehicle states based on MCSCKF under non-Gaussian noise using the obtained mass and tire force information.(2)To verify the effectiveness of the proposed method, two typical test scenarios are performed. The results demonstrate that the proposed robust hierarchical estimation scheme can accurately estimate the vehicle mass, tire force, and vehicle driving state. Moreover, the MCSCKF has better accuracy and robustness for vehicle state estimation in non-Gaussian situations compared to the conventional Kalman filter.

The remainder of the paper is presented below. Section 2 describes the adopted vehicle model. The robust hierarchical estimation scheme is presented in Section 3. The validity of the proposed method is verified in Section 4. Section 5 summarizes this work.

## 2. Vehicle Model

To accurately identify the vehicle mass, a lateral dynamics model is initially used to estimate the vehicle mass. Then, a single-track model and a three-degree-of-freedom nonlinear dynamics model are used to estimate the vehicle’s tire force and driving state, respectively. In addition, the calculation of the vertical tire force is also necessary.

For vehicle state estimation systems, some assumptions are necessary:The vertical motion of the vehicle along the Z-axis is constant, and the effect of the suspension is ignored.The vehicle dynamics model ignores the influence of the steering system, and the front-wheel angle is directly used as the model input.During the operation of the vehicle, the roll motion of the X-axis and the pitch motion of the Y-axis are ignored.Ignoring air and wind resistance, the vehicle is mainly subjected to tire-road forces.

### 2.1. Vehicle Mass Estimation Model

The estimation model of the vehicle mass [3] is shown in Equation (1). Generally, the identification of the mass is performed at the start of the vehicle, when the tire is in the linear region and the cornering stiffnesses $C_{f}$ and $C_{r}$ are considered to be a constant value.
(1)$$ma_{y}=\frac{(l_{f}C_{f}-l_{r}C_{r})r}{v_{x}}+\beta (C_{f}+C_{r})-C_{f}\delta$$
where m is the vehicle mass, $a_{y}$ denotes the lateral acceleration, $v_{x}$ is the longitudinal vehicle speed, r represents the yaw rate, β is the sideslip angle, δ is the front wheel angle, $l_{f}$ and $l_{r}$ denote the distance from the front and rear axles to the center of gravity (CG), and $C_{f}$ and $C_{r}$ are the cornering stiffnesses of the front and rear wheels, respectively.

### 2.2. Tire Force Estimation Model

For the estimation of longitudinal tire forces, the vehicle single-wheel rolling dynamics model [28] is employed herein, as shown in Figure 1.

From Figure 1, the rotational dynamics equation for each wheel is
(2)$$J{\dot{\omega }}_{ij}=-R_{e}F_{xij}+T_{dij},\;i=1,2,3,4\;j=1,2,3,4$$
where J is the moment of inertia of the wheel, $R_{e}$ is the effective radius of the tire, $\omega_{ij}$ is the rotational angular velocity of the wheel, $F_{xij}$ is the longitudinal force, and $T_{dij}$ is the driving torque of the tire. In the whole article, ij = 11, 12, 21, and 22, representing the left front, right front, left rear, and right rear wheels, respectively. $\omega_{ij}$ can be obtained from the on-board sensors, but the direct calculation for the longitudinal tire force using Equation (2) introduces noise differential error, which has a large impact on the calculation results. Therefore, the estimation of the longitudinal tire force becomes an optional solution.

The single-track model, also known as the bicycle model, is widely used in vehicle dynamics and stability control [29]. This model is utilized to design the vehicle lateral tire force observer, and Figure 2 illustrates the single-track plane model with the mathematical equation as in Equation (3).
(3)$$\{\begin{array}{l} {\dot{v}}_{y}=\left(F_{yf}\cos\delta +F_{xf}\sin\delta +F_{yr}\right)/m\\ \dot{r}=\left(l_{f}F_{yf}\cos\delta +l_{f}F_{xf}\sin\delta -l_{r}F_{yr}\right)/I_{z} \end{array}$$
where $v_{y}$ denotes the lateral vehicle speed along the Y-axis and $I_{z}$ is the vehicle moment of inertia in the X-Y plane. $F_{xf}$ and $F_{yf}$ denote the total longitudinal and lateral tire forces at the front wheels, and $F_{yr}$ is the total lateral tire force at the rear wheels.

### 2.3. Nonlinear Three-Degree-of-Freedom Dynamics Model

A simplified three-degree-of-freedom, four-wheeled vehicle model [9] is shown in Figure 3. The nonlinear dynamics equations containing longitudinal, lateral, and yaw motions are as follows
(4)$$m({\dot{v}}_{x}-v_{y}r)=(F_{x11}+F_{x12})\cos\delta -(F_{y11}+F_{y12})\sin\delta +F_{x21}+F_{x22}$$
(5)$$m({\dot{v}}_{y}+v_{x}r)=(F_{x11}+F_{x12})\sin\delta +(F_{y11}+F_{y12})\cos\delta +F_{y21}+F_{y22}$$
(6)$$\begin{array}{l} I_{z}\dot{r}=[(F_{x11}+F_{x12})\sin\delta +(F_{y11}+F_{y12})\cos\delta ]l_{f}+[(F_{x11}+F_{x12})\cos\delta +(F_{y11}-F_{y12})\sin\delta ]\frac{t_{w1}}{2}\\ +(F_{x21}+F_{x22})\frac{t_{w2}}{2}-(F_{y21}+F_{y22})l_{r}] \end{array}$$
where $F_{xij}$ and $F_{yij}$ (ij=11,12,21,22) represent the longitudinal and lateral tire forces, respectively. $t_{w1}$ and $t_{w2}$ are the front and rear track widths, and the meanings of the remaining symbols are the same as those mentioned earlier. From Figure 3, the sideslip angle $\beta =\tan^{-1}\left(\frac{v_{y}}{v_{x}}\right)$.

### 2.4. Tire Vertical Force Calculation

When the vehicle has significant acceleration, deceleration, and turning, the tire vertical force will be redistributed due to load transfer [23]. Considering the effects of vehicle acceleration, deceleration, and lateral motion, the tire vertical force is calculated as
(7)$$\begin{array}{l} F_{z11}=m_{w}g+m_{s}g\frac{l_{r}}{2(l_{f}+l_{r})}-m_{s}a_{x}\frac{h_{g}}{2(l_{f}+l_{r})}-m_{s}a_{y}\frac{h_{g}}{2t_{w1}}\\ F_{z12}=m_{w}g+m_{s}g\frac{l_{r}}{2(l_{f}+l_{r})}-m_{s}a_{x}\frac{h_{g}}{2(l_{f}+l_{r})}+m_{s}a_{y}\frac{h_{g}}{2t_{w1}}\\ F_{z21}=m_{w}g+m_{s}g\frac{l_{f}}{2(l_{f}+l_{r})}+m_{s}a_{x}\frac{h_{g}}{2(l_{f}+l_{r})}-m_{s}a_{y}\frac{h_{g}}{2t_{w2}}\\ F_{z22}=m_{w}g+m_{s}g\frac{l_{f}}{2(l_{f}+l_{r})}+m_{s}a_{x}\frac{h_{g}}{2(l_{f}+l_{r})}+m_{s}a_{y}\frac{h_{g}}{2t_{w2}} \end{array}$$
where $F_{zij}$ denotes the vertical tire force, $m_{w}$ is the unsprung mass, $m_{s}$ is the sprung mass, $a_{x}$ is the longitudinal acceleration, $h_{g}$ is the height of the CG, and *g* is the gravitational acceleration.

## 3. Robust Hierarchical Estimation Scheme

Considering that the vehicle mass may change when the vehicle starts or stops, such as when a car turns from empty to full, the vehicle’s mass is identified first. The accuracy of tire force calculations is crucial for vehicle dynamics control. However, common empirical (semi-empirical) and theoretical tire models require several tire characteristic parameters that are not easily obtained accurately and have time-varying characteristics, resulting in the inaccurate output of formula-based tire models. Meanwhile, the on-board sensor signals are susceptible to non-Gaussian noise (e.g., heavy-tailed noise with large outliers) when the vehicle is in actual operation, which will affect the accuracy and robustness of vehicle state estimation. To address the above issues, a robust hierarchical estimation scheme is proposed, as shown in Figure 4. The sensor measurement module can provide easily measured on-board sensor signals, including steering wheel angle, wheel angular velocity, inertial sensor signals, wheel speed signals, etc. Using the FRLS method, the vehicle mass is first accurately estimated by the vehicle mass identification module. Secondly, longitudinal and lateral tire forces are further estimated using ASMO. Finally, the vehicle driving state is estimated based on MCSCKF using the estimated mass and tire forces. It is worth noting that the information in the whole estimation scheme is interchangeable.

### 3.1. Vehicle Mass Identification

Recursive least squares (RLS) is computationally small and converges quickly, and to make full use of real-time measurement information, FRLS is used to estimate the mass of a vehicle [15]. The general recursive form of RLS is
(8)$$y(k)=\varphi^{T}(k)\theta +e(k)$$
where y(k) is the system output, θ and $\varphi^{T}(k)$ are the estimated parameter and recursive vector, and e(k) is the error term. Combining with Equation (1), the regression model for vehicle mass estimation is
(9)$$\left\{ \begin{array}{l} \varphi^{T}(k)=a_{y}\;\theta =m\\ y(k)=\frac{(l_{f}C_{f}-l_{r}C_{r})r}{v_{x}}+\beta (C_{f}+C_{r})-C_{f}\delta \end{array}\right.$$

The main recursive steps of the FRLS algorithm are as follows.

The gain $K_{F}$ is calculated
(10)$$K_{F}(k)=\frac{P_{F}(k-1)\varphi (k)}{\lambda +\varphi^{T}(k)P_{F}(k-1)\varphi (k)}$$

The estimated parameter is updated
(11)$$\theta (k)=\theta (k-1)+K_{F}(k)[y(k)-\varphi^{T}(k)\theta (k-1)]$$

The covariance matrix is updated
(12)$$P_{F}(k)=\frac{1}{\lambda }[I-K_{F}(k)\varphi^{T}(k)]P_{F}(k-1)$$
where I is the unit matrix and λ is the forgetting factor, which usually takes the value in the interval [0.9, 1], in this paper, λ is set to 0.99.

### 3.2. Tire Force Estimation

#### 3.2.1. Sliding Mode Observer Design

Sliding mode control is a design method for automatic control systems with strong robustness to system parameter uncertainties and external disturbances [30]. The designed state observer based on sliding mode control theory is also robust and has excellent performance. In this paper, an adaptive sliding mode observer is designed to estimate the longitudinal and lateral tire forces using easily measurable on-board sensor information.

Consider a first-order system
(13)$$\left\{ \begin{array}{l} {\dot{x}}_{s}=B_{s}u_{s}+P_{s}d+\psi_{s}\\ y_{s}=x_{s} \end{array}\right.$$
where $x_{s}$ is the system state, $u_{s}$ is the system input, d is the unknown and bounded input, $y_{s}$ is the measured output, $\psi_{s}$ is the disturbance term, $B_{s}$ and $P_{s}$ are the real constants, respectively. When the system’s state changes, the unknown input d varies as well, where d is the state to be estimated.

Define the error of the system as ${\tilde{x}}_{s}=x_{s}-{\hat{x}}_{s}$. In this paper, the sliding mode surface is chosen as the system error, that is, $S={\tilde{x}}_{s}$. Then the Lyapunov function is designed as [31]
(14)$$V=S^{2}/2$$

Then, the derivative of Equation (14) is
(15)$$\dot{V}=S\dot{S}=S{\dot{\tilde{x}}}_{s}=S({\dot{x}}_{s}-{\dot{\hat{x}}}_{s})$$

According to the state observer theory, based on Equation (13), an observer can be constructed
(16)$${\dot{\hat{x}}}_{s}=B_{s}u_{s}+P_{s}\hat{d}+\psi_{s}+L(x_{s}-{\hat{x}}_{s})$$
where L is the observer gain.

Taking Equations (13) and (16) into Equation (15), we have
(17)$$\dot{V}=S[(B_{s}u_{s}+P_{s}d+\psi_{s})-(B_{s}u_{s}+P_{s}\hat{d}+\psi_{s}+L(x_{s}-{\hat{x}}_{s}))]=-SP_{s}\hat{d}+S[P_{s}d-L(x_{s}-{\hat{x}}_{s})]$$

Since d is an unknown and bounded input, there always exists a positive real number ρ, and when ρ is large enough, the following inequality holds.
(18)$$\left|P_{s}d-L(x_{s}-{\hat{x}}_{s})\right|⩽\rho$$

Substituting Equation (18) into Equation (17), the following equation can be obtained
(19)$$\begin{array}{l} \dot{V}=-SP_{s}\hat{d}+S[P_{s}d-L(x_{s}-{\hat{x}}_{s})]\\ ⩽-SP_{s}\hat{d}+S\rho \\ ⩽-SP_{s}\hat{d}+\left|S\right|\rho \end{array}$$

The following definition is made
(20)$$\hat{d}=P_{s}{}^{-1}\rho \mathrm{sgn}(S)$$
(21)$$\mathrm{sgn}(S)=\{\begin{array}{c} 1,\;S>0\\ 0,\;S=0\\ -1,\;S<0 \end{array}$$
where sgn(S) denotes the signum function.

Thus, Equation (19) can be further written as
(22)$$\dot{V}⩽-SP_{s}\hat{d}+\left|S\right|\rho =-S\rho \mathrm{sgn}(S)+\left|S\right|\rho =0$$

When the derivative of the Lyapunov function satisfies $\dot{V}\leq 0$, the system is stable, and the state variables can converge to the sliding mode surface [5]. Thus, we can obtain the state observer depicted as follows
(23)$$\left\{ \begin{array}{l} {\dot{\hat{x}}}_{s}=B_{s}u_{s}+P_{s}\hat{d}+\psi_{s}+L(x_{s}-{\hat{x}}_{s})\\ \hat{d}=P_{s}{}^{-1}\rho \mathrm{sgn}(y_{s}-{\hat{x}}_{s}) \end{array}\right.$$

Combining Equations (13) and (23), the error derivative of the system is further expressed as
(24)$$\begin{array}{cc} {\dot{\tilde{x}}}_{s} & =({\dot{x}}_{s}-{\dot{\hat{x}}}_{s})\\ & =B_{s}u_{s}+P_{s}d+\psi_{s}-[B_{s}u_{s}+\rho \mathrm{sgn}(y_{s}-{\hat{x}}_{s})+\psi_{s}+L(x_{s}-{\hat{x}}_{s})]\\ & =P_{s}d-\rho \mathrm{sgn}(y_{s}-{\hat{x}}_{s})-L(x_{s}-{\hat{x}}_{s}) \end{array}$$

When the system reaches stability, the state variables converge to the sliding surface, ${\dot{\tilde{x}}}_{s}=0$, and based on Equation (24), the observer for the unknown input d, i.e., $F_{yf}$ and $F_{yr}$ can be expressed as
(25)$$\hat{d}\_z=P_{s}{}^{-1}\rho \mathrm{sgn}(y_{s}-{\hat{x}}_{s})+P_{s}{}^{-1}L(x_{s}-{\hat{x}}_{s})$$
where $\hat{d}\_z$ denotes the final estimate of the unknown input d. L is the feedback gain, and ρ is the sliding mode gain.

Due to the effects of time and space delays and system inertia, the sliding mode system is prone to trembling, which will increase the estimation error and affect the estimation results. To eliminate the effect of trembling, the saturation function sat(S) is used to replace the signum function sgn(S) [5].
(26)$$sat(S)=\{\begin{array}{l} \frac{S}{\kappa },\;\left|S\right|\leq \kappa \\ \mathrm{sgn}(S),\left|S\right|>\kappa \end{array}$$
where κ>0 and can adjust the slope of the function curve; in this paper, κ=0.01.

#### 3.2.2. Longitudinal Tire Force Estimation

According to Equation (13), Equation (2) can be transformed into
(27)$$\{\begin{array}{l} {\dot{\omega }}_{ij}=\frac{1}{J}T_{dij}-\frac{R_{e}}{J}F_{xij}\\ y=\omega_{i} \end{array}$$
where $\omega_{ij}$ is both the state variable and the measured output of the system, $T_{dij}$ is the system input, and $F_{xij}$ is the variable to be estimated.

From Equations (25) and (26), the longitudinal tire force can be estimated as
(28)$${\hat{F}}_{xij}=\frac{J}{R_{e}}\rho_{xij}sat(\omega_{ij}-{\hat{\omega }}_{ij})+\frac{J}{R_{e}}L_{xij}(\omega_{ij}-{\hat{\omega }}_{ij})$$
where ${\hat{\omega }}_{ij}$ is the estimate of $\omega_{ij}$ and ${\hat{\omega }}_{ij}$ can be obtained from ${\dot{\omega }}_{ij}=\frac{1}{J}T_{dij}-\frac{R_{e}}{J}F_{xij}$ by the first-order Eulerian discretization method, i.e., that is $\omega_{ij\_k+1}=\omega_{ij\_k}(\frac{1}{J}T_{dij}-\frac{R_{e}}{J}F_{xij})T$, where the subscript *k* denotes the sampling moment and *T* is the sampling time. $L_{xij}(ij=11,12,21,22)$ is the longitudinal tire force sliding mode observer feedback gain, and $L_{x11}=L_{x12}=L_{x21}=L_{x22}=200$. $\rho_{xij}$ is the sliding mode gain, and $\rho_{x11}=\rho_{x12}=\rho_{x21}=\rho_{x22}=10$.

#### 3.2.3. Lateral Tire Force Estimation

Decoupling $F_{yf}$ and $F_{yr}$ in Equation (3), we have
(29)$$\left\{ \begin{array}{l} F_{yf}=[I_{z}\dot{r}+l_{r}ma_{y}-F_{xf}\sin\delta (l_{f}+l_{r})]/[(l_{f}+l_{r})\cos\delta ]\\ F_{yr}=(l_{f}ma_{y}-I_{z}\dot{r})/(l_{f}+l_{r}) \end{array}\right.$$

Similar to the principle of longitudinal tire force estimation from Equations (13), (25), and (26), the lateral tire force estimation is given by
(30)$${\hat{F}}_{yf}=\frac{I_{z}}{\cos\delta (l_{f}+l_{r})}\rho_{yf}sat(r-\hat{r})+\frac{I_{z}}{\cos\delta (l_{f}+l_{r})}L_{yf}(r-\hat{r})\;{\hat{F}}_{yr}=\frac{-I_{z}}{\left(l_{f}+l_{r}\right)}\rho_{yr}sat(r-\hat{r})-\frac{I_{z}}{\left(l_{f}+l_{r}\right)}L_{yr}(r-\hat{r})$$
where $\hat{r}$ is the estimate of r and $\hat{r}$ can be obtained from $\dot{r}=-\frac{l_{r}m}{I_{z}}a_{y}+\frac{\left(l_{f}+l_{r}\right)\cos\delta }{I_{z}}F_{yf}+\frac{\left(l_{f}+l_{r}\right)\sin\delta }{I_{z}}F_{xf}$ by the first-order Eulerian discretization method, i.e., that is $r_{\_k+1}=r_{\_k}+(-\frac{l_{r}m}{I_{z}}a_{y}+\frac{(l_{f}+l_{r})\cos\delta }{I_{z}}F_{yf}+\frac{(l_{f}+l_{r})\sin\delta }{I_{z}}F_{xf})T$. $L_{yf}$ and $L_{yr}$ are the feedback gains of the front and rear wheel lateral force sliding mode observers, respectively, and $L_{yf}=L_{yr}=20$. $\rho_{yf}$ and $\rho_{yr}$ are the sliding mode gains, and $\rho_{yf}=\rho_{yr}=0.1$, respectively.

An adaptive feedback law is designed to improve the estimation accuracy of $F_{yf}$ and $F_{yr}$. The adaptive sliding mode observer is
(31)$${\hat{F}}_{yf,ada}=\frac{I_{z}}{\cos\delta (l_{f}+l_{r})}\rho_{yf}sat(r-\hat{r})+\frac{I_{z}}{\cos\delta (l_{f}+l_{r})}L_{yf}(r-\hat{r})+\eta_{yf}(ma_{y}-{\hat{F}}_{yf}-{\hat{F}}_{yr})\;{\hat{F}}_{yr,ada}=\frac{-I_{z}}{\left(l_{f}+l_{r}\right)}\rho_{yr}sat(r-\hat{r})-\frac{I_{z}}{\left(l_{f}+l_{r}\right)}L_{yr}(r-\hat{r})+\eta_{yr}(ma_{y}-{\hat{F}}_{yf}-{\hat{F}}_{yr})$$
where ${\hat{F}}_{yf,ada}$ and ${\hat{F}}_{yr,ada}$ indicate the final estimated longitudinal and lateral tire forces. $\eta_{yf}$ and $\eta_{yr}$ are the adaptive observer feedback gains, and $\eta_{yf}=\eta_{yr}=1$, respectively.

Finally, the lateral tire force for each tire is estimated as
(32)$$\{\begin{array}{l} {\hat{F}}_{y11}=\frac{F_{z11}}{F_{z11}+F_{z12}}{\hat{F}}_{yf}\\ {\hat{F}}_{y12}=\frac{F_{z12}}{F_{z11}+F_{z12}}{\hat{F}}_{yf}\\ {\hat{F}}_{y21}=\frac{F_{z21}}{F_{221}+F_{z22}}{\hat{F}}_{yr}\\ {\hat{F}}_{y22}=\frac{F_{z22}}{F_{z21}+F_{z22}}{\hat{F}}_{yr} \end{array}$$

### 3.3. Vehicle State Estimation Based on MCSCKF

The EKF is used for nonlinear systems by linearization, but it has only first-order accuracy and large estimation errors for high-dimensional nonlinear systems. UKF has higher accuracy than EKF, but UKF often faces the problem of covariance matrix non-positive definite, which leads to estimation divergence. Later, the cubature Kalman filter (CKF) was proposed by scholars. According to the spherical-radial cubature rule, the CKF can calculate multivariate moment integrals in the nonlinear Bayesian filtering scheme, which enables the CKF to have better filtering performance in high-dimensional nonlinear systems. The SCKF is a square-root version of the CKF, and to prevent the asymmetry and negative definiteness of the covariance matrix, the SCKF uses least squares and triangulation to update the Kalman gain and covariance matrices. This not only enhances the robustness of the numerical computation but also ensures the positive definiteness of the covariance matrix, which can effectively improve the accuracy of the estimation [9]. The conventional Kalman filter is based on the MMSE criterion and assumes that the noise is Gaussian distributed, which has good performance in a Gaussian environment. However, in actual vehicle operation, the sensor signals are vulnerable to large outliers, and the noises are often heavy-tailed with non-Gaussian characteristics, etc., which degrades the performance of the conventional Kalman filter [32]. MCC can capture the second- and higher-order moments of the errors and has a strong suppression effect on non-Gaussian noises [24]. In this paper, the MCSCKF algorithm is derived by combining the MCC and SCKF, and it is applied to the vehicle driving state estimation in real-world operating conditions. The design of the state estimator and the MCSCKF algorithm will be presented in the following.

#### 3.3.1. Design of the State Estimator

Generally, the discrete state and measurement equations of a nonlinear system can be described as
(33)x(k)=f(x(k−1),u(k−1))+v(k−1)
(34)z(k)=h(x(k),u(k))+w(k)
where x(k)∈ℝn and z(k)∈ℝm denote the n-dimensional state vector and the m-dimensional measurement vector, respectively. f(·) and h(·) are the nonlinear state and measurement functions of the system, and u(k) is the system input. v(k−1) and w(k) represent the uncorrelated process and measurement noises with zero means and the covariance matrix Q(k−1)=E[v(k−1)vT(k−1)], R(k)=E[w(k)wT(k)].

Based on Equations (4)–(6), employing the first-order Euler discrete method, the state and measurement equations of the vehicle system can be expressed as
(35)$$\begin{array}{l} \left[\begin{array}{l} v_{x}(k)\\ v_{y}(k)\\ r\;(k) \end{array}\right]=\left[\begin{array}{l} v_{x}(k-1)\\ v_{y}(k-1)\\ r\;(k-1) \end{array}\right]\\ =\left[\begin{array}{c} \left(\begin{array}{l} \left[(F_{x11}(k-1)+F_{x12}(k-1))\cos\delta (k-1)-(F_{y11}(k-1)+F_{y12}(k-1))\sin\delta (k-1\right)\\ +F_{x21}(k-1)+F_{x22}(k-1)+m(k-1)v_{y}(k-1)r(k-1)]/m(k-1) \end{array}\right)\\ \left(\begin{array}{l} \left[(F_{x11}(k-1)+F_{x12}(k-1))\sin\delta (k-1)+(F_{y11}(k-1)+F_{y12}(k-1))\cos\delta (k-1\right)\\ +F_{y21}(k-1)+F_{y22}(k-1)-m(k-1)v_{x}(k-1)r(k-1)]/m(k-1) \end{array}\right)\\ \left(\begin{array}{l} {}[(F_{x11}(k-1)+F_{x12}(k-1))\sin\delta (k-1)+(F_{y11}(k-1)+F_{y12}(k-1))\cos\delta (k-1)]l_{f}\\ +[(F_{x11}(k-1)+F_{x12}(k-1))\cos\delta (k-1)+(F_{y11}(k-1)-F_{y12}(k-1))\sin\delta (k-1)]\frac{t_{w1}}{2}\\ +(F_{x21}(k-1)+F_{x22}(k-1))\frac{t_{w2}}{2}-(F_{y21}(k-1)+F_{y22}(k-1))l_{r}]/I_{z} \end{array}\right) \end{array}\right]T+v_{k-1} \end{array}$$
where *T* is the sampling time of the system.

Modern production cars are equipped with a large number of on-board sensors required for the ESC. For example, steering wheel angle sensors (front wheel angle δ can be obtained by a ratio coefficient), wheel speed sensors (four wheels speed $v_{cij}$, ij=11,12,21,22), accelerometers (longitudinal and lateral acceleration $a_{x}$ and $a_{y}$) and gyroscopes (yaw rate r) [4]. Also, these sensors can share data with the vehicle controller area network (CAN) bus. Utilizing the above easily available on-board sensor information, the measurement equations of the system are designed as
(36)$$\left[\begin{array}{l} a_{x,m}(k)\\ a_{y,m}(k)\\ r_{,m}(k)\\ v_{c11,m}(k)\\ v_{c12,m}(k)\\ v_{c21,m}(k)\\ v_{c22,m}(k) \end{array}\right]=\left[\begin{array}{c} \left(\begin{array}{l} \left[(F_{x11}(k)+F_{x12}(k))\cos\delta (k)-(F_{y11}(k)+F_{y12}(k))\sin\delta (k\right)\\ +F_{x21}(k)+F_{x22}(k)+m(k)v_{y}(k)r(k)]/m(k) \end{array}\right)\\ \left(\begin{array}{l} \left[(F_{x11}(k)+F_{x12}(k))\sin\delta (k)+(F_{y11}(k)+F_{y12}(k))\cos\delta (k\right)\\ +F_{y21}(k)+F_{y22}(k)-m(k)v_{x}(k)r(k)]/m(k) \end{array}\right)\\ r(k)\\ \left(v_{x}(k)-\frac{t_{w1}}{2}r(k))\cos\delta (k)+(v_{y}(k)+l_{f}r(k))\sin\delta (k\right)\\ \left(v_{x}(k)+\frac{t_{w1}}{2}r(k))\cos\delta (k)+(v_{y}(k)+l_{f}r(k))\sin\delta (k\right)\\ v_{x}(k)-\frac{t_{w2}}{2}r(k)\\ v_{x}(k)+\frac{t_{w2}}{2}r(k) \end{array}\right]+w_{k}$$
where the subscript m indicates the sensor measurement.

From Equations (35) and (36), the state and measurement vectors of the vehicle state estimator are
(37)$$x(k)={\left[v_{x},v_{y},r\right]}^{T}$$
(38)$$z(k)={\left[a_{x,m},a_{y,m},r_{m},v_{c11,m},v_{c12,m},v_{c21,m},v_{c22,m}\right]}^{T}$$

#### 3.3.2. Maximum Correntropy Square-Root Cubature Kalman Filter

(1)Maximum correntropy criterion

The correntropy is a nonlinear measure for two random variables X,Y∈ℝ, and assuming that their joint distribution function is $F_{XY}(x,y)$, the correntropy can be expressed as [24]
(39)$$V(X,Y)=E[\kappa (X,Y)]=\int \kappa (x,y)dF_{XY}(x,y)$$
where E[·] denotes the expectation function and κ(· , ·) is the kernel function. Usually, the Gaussian kernel is chosen as the kernel function.
(40)$$\kappa (x,y)=G_{\sigma }(e)=\exp(-\frac{e^{2}}{2\sigma^{2}})$$
where e=x−y, σ>0 denotes the kernel width of the correntropy.

In practical situations, the joint distribution function of the variables $F_{XY}(x,y)$ is often unknown, and limited sample data ${\left\{x(i),y(i)\right\}}_{i=1}^{N}$ is available, so the correntropy is often estimated as
(41)$$\hat{V}(X,Y)=\frac{1}{N}\sum_{i=1}^{N}G_{\sigma }(e(i))$$
where e(i)=x(i)−y(i).

Allowing the correntropy as an objective function has a strong suppression effect on non-Gaussian heavy-tailed noise [25]. If the error data ${\left\{e(i)\right\}}_{i=1}^{N}$ can be obtained, the MCC-based objective function is expressed as
(42)$$J_{MCC}=\frac{1}{N}\sum_{i=1}^{N}G_{\sigma }(e(i))$$

(2)Square-root cubature Kalman filter

For the state estimation of high-dimensional systems, SCKF is a powerful tool. Taking the nonlinear system in Equations (33) and (34) as an example, the SCKF has two main steps, prediction and update.

Predict

Assume that at some time k, S(k−1|k−1) is the square-root of the state error covariance matrix P(k−1|k−1), that is, $P(k-1|k-1)=S(k-1|k-1)S^{T}(k-1|k-1)$. Similarly, $S_{Q}(k-1)$ and $S_{R}(k)$ are the square-root factors of Q(k−1) and R(k), that is, $Q(k-1)=S_{Q}(k-1)S_{Q}{}^{T}(k-1)$, $R(k)=S_{R}(k)S_{R}{}^{T}(k)$.

Calculation of cubature points
(43)$$\chi^{i}(k-1|k-1)=S(k-1|k-1)\cdot I(i)+\hat{x}(k-1|k-1),i=1,\ldots ,2n$$
with $I(i)=\left\{ \begin{array}{ll} \sqrt{n}{\left[1\right]}_{i} & \;i=1,\ldots ,n\\ -\sqrt{n}{\left[1\right]}_{i-n} & \;i=n+1,\ldots ,2n \end{array}\right.$
where I is the unit matrix of n×n and ${\left[1\right]}_{i}$ is the *i*-th column vector.

Propagation of cubature points
(44)$$\chi^{i*}(k|k-1)=f(k-1,\chi^{i}(k-1|k-1)),i=1,\ldots ,2n$$

The prior state and the square root of the covariance matrix are estimated by
(45)$$\hat{x}(k|k-1)=\frac{1}{2n}\sum_{i=1}^{2n}\chi^{i*}(k|k-1)$$
(46)$$S(k|k-1)=Tria([X^{*}(k|k-1),S_{Q}(k-1)])$$
with $X^{*}(k|k-1)=\frac{1}{\sqrt{2n}}[\chi^{1*}(k|k-1)-\hat{x}(k|k-1),\ldots ,\chi^{2n*}(k|k-1)-\hat{x}(k|k-1)],i=1,\ldots ,2n$. Tria(·) represents the triangular decomposition of the matrix.

2.Update

Calculation of cubature points
(47)$$\chi^{i}(k|k-1)=S(k|k-1)\cdot I(i)+\hat{x}(k|k-1),i=1,\ldots ,2n$$

Propagation of cubature points
(48)$$\chi^{i**}(k|k-1)=h(k,\chi^{i}(k|k-1)),i=1,\ldots ,2n$$

The prior measurement and the square root of the covariance matrix are estimated by
(49)$$\hat{z}(k|k-1)=\frac{1}{2n}\sum_{i=1}^{2n}\chi^{i**}(k|k-1)$$
(50)$$S_{zz}(k|k-1)=Tria([Z(k|k-1),S_{R}(k)])$$
with $Z(k|k-1)=\frac{1}{\sqrt{2n}}[\chi^{1**}(k|k-1)-\hat{z}(k|k-1),\cdots ,\chi^{2n**}(k|k-1)-\hat{z}(k|k-1)],i=1,\ldots ,2n$.

Calculate the cross-covariance matrix
(51)$$P_{xz}(k|k-1)=X(k|k-1)Z^{T}(k|k-1)$$
with $X(k|k-1)=\frac{1}{\sqrt{2n}}[\chi^{1}(k|k-1)-\hat{x}(k|k-1),\ldots ,\chi^{2n}(k|k-1)-\hat{x}(k|k-1)],i=1,\ldots ,2n$.

Calculate the Kalman gain
(52)$$K(k)=P_{xz}(k|k-1)/S_{zz}^{T}(k|k-1)/S_{zz}(k|k-1)$$

The posterior state and the square root of the covariance matrix are estimated by
(53)$$\hat{x}(k|k)=\hat{x}(k|k-1)+K(k)(z(k)-\hat{z}(k|k-1))$$
(54)$$S(k|k)=Tria([X(k|k-1)-K(k)Z(k|k-1),K(k)S_{R}(k)])$$

3.Derivation of the MCSCKF

Since the correntropy has excellent performance in the non-Gaussian environment, we combine MCC and SCKF to derive the MCSCKF algorithm, and the detailed derivation process is as follows.

For the nonlinear model described in Equations (33) and (34), combined with Equations (45) and (46), a nonlinear recursive model is constructed as
(55)$$\left[\begin{array}{c} \hat{x}(k|k-1)\\ z(k) \end{array}\right]=\left[\begin{array}{c} x(k)\\ h(k,x(k)) \end{array}\right]+\phi (k)$$
where $\phi (k)=\left[\begin{array}{c} \hat{x}(k|k-1)-x(k)\\ w(k) \end{array}\right]$, the covariance of the matrix ϕ(k) can be written $\begin{array}{l} E(\phi_{k}\phi_{k}^{T})=\left[\begin{array}{cc} P(k|k-1) & 0\\ 0 & R(k) \end{array}\right]=\left[\begin{array}{cc} M_{p}(k|k-1)M_{p}^{T}(k|k-1) & 0\\ 0 & M_{r}(k)M_{r}^{T}(k) \end{array}\right]\\ =M(k)M^{T}(k) \end{array}$. M(k) is obtained from the Cholesky decomposition of $E(\phi_{k}\phi_{k}^{T})$.

Multiplying $M^{-1}(k)$ left on both sides of Equation (55) yields
(56)D(k)=g(k,x(k))+e(k)
where $D(k)=M^{-1}(k)\left[\begin{array}{c} \hat{x}(k|k-1)\\ z(k) \end{array}\right],g(k,x(k))=M^{-1}(k)\left[\begin{array}{c} x(k)\\ h(k,x(k)) \end{array}\right]$ and $e(k)=M^{-1}(k)\phi (k)$.

Then, based on MCC, we define the following objective function
(57)$$J_{MCC}(x(k))=\sum_{i=1}^{n+m}G_{\sigma }(e_{i}(k))=\sum_{i=1}^{n+m}G_{\sigma }(d_{i}(k)-g_{i}(k,x(k)))$$
where $d_{i}(k)$ is the *i*-th element of D(k) and $g_{i}(k,x(k))$ is the *i*-th row of g(k,x(k)).

Then, the optimal estimate of x(k) based on MCC can be obtained from the following equation
(58)$$\hat{x}(k)=\arg\underset{x(k)}{\max}\sum_{i=1}^{n+m}G_{\sigma }(e_{i}(k))$$

Let the first-order derivative of Equation (58) be zero, and the optimal solution of (58) can be obtained.
(59)$$\frac{\partial J_{MCC}(x(k))}{\partial x(k)}=\sum_{i=1}^{n+m}G_{\sigma }(e_{i}(k))\cdot e_{i}(k)\cdot \frac{\partial e_{i}(k)}{\partial x(k)}=0$$

Then, defining $C_{i}(k)=G_{\sigma }(e_{i}(k))$, we have
(60)$$C(k)=\mathrm{diag}(C_{1}(k),\ldots ,C_{n+m}(k))=\left[\begin{array}{cc} C_{\left(x\right)}(k) & 0\\ 0 & C_{\left(y\right)}(k) \end{array}\right]$$
with $C_{\left(x\right)}(k)={\mathrm{diag}(G}_{\sigma }(e_{1}(k)),\ldots ,G_{\sigma }(e_{n}(k)))$ and $C_{\left(y\right)}(k)={\mathrm{diag}(G}_{\sigma }(e_{n+1}(k)),\ldots ,G_{\sigma }(e_{n+m}(k)))$.

Based on Equation (60), Equation (59) can be further expressed as
(61)$${\left(\frac{\partial g(k,x(k))}{\partial x(k)}\right)}^{T}C(k)(D(k)-g(k,x(k)))=0$$

Actually, the key to using MCC to improve the performance of SCKF in a non-Gaussian environment is using C(k) to update the state covariance and the variance of measurement noise [26]. We define the updated covariance matrix of ϕ(k) as $\tilde{L}(k)$
(62)$$\tilde{L}(k)=\left[\begin{array}{cc} \tilde{P}(k|k-1) & 0\\ 0 & \tilde{R}(k) \end{array}\right]=M(k)\cdot C{\left(k\right)}^{-1}\cdot M^{T}(k)$$

In practice, the true state x(k) is often unknown, let $\hat{x}(k|k-1)-x(k)=0$. Therefore, the prior measurement noise variance are written as
(63)$$\tilde{R}(k)=M_{r}(k)C_{\left(y\right)}^{-1}(k)M_{r}^{T}(k)$$

Then, the square root $S_{\tilde{R}}(k)$ of the updated measurement covariance matrix can be obtained from
(64)$$S_{\tilde{R}}(k)={\left(M_{r}(k)C_{\left(y\right)}(k)\right)}^{-1/2}$$

Finally, combining the above derivation with the prior estimation process of SCKF (Equations (43)–(46)), the derivation of MCSCKF is completed, and the main steps are summarized in the Algorithm 1.

**Algorithm 1:** MCSCKF
**1 Input**
*σ*, a positive number
ε$,\;\hat{x}(0|0)$, S(0|0)
**2 Initialization**
*k* = 1
**3 Time Update**
 for *i* = 1, …, 2*n*  $\chi^{i}(k-1|k-1)=S(k-1|k-1)\cdot I(i)+\hat{x}(k-1|k-1),i=1,\ldots ,2n$  $\chi^{i*}(k|k-1)=f(k-1,\chi^{i}(k-1|k-1)),i=1,\ldots ,2n$ end $\hat{x}(k|k-1)=\frac{1}{2n}\sum_{i=1}^{2n}\chi^{i*}(k|k-1)$ $S(k|k-1)=Tria([X^{*}(k|k-1),S_{Q}(k-1)])$
**4 Measurement Update**
 for *i* = 1, …, 2*n*  χi(k|k−1)=S(k|k−1)·I(i)+x^(k|k−1),i=1,…,2n  χi∗∗(k|k−1)=h(k,χi(k|k−1)),i=1,…,2n end z^(k|k−1)=12n∑i=12nχi∗∗(k|k−1)4.1 Initialization: *t* = 1, x^(k|k)(0)=x^(k|k−1)Mr(k)=chol(R(k))4.2 Iteration: ej(t−1)(k)=dj(k)−gj(k,x^(k|k)(t−1)),j=1,…,n+m C(y)(t−1)(k)=diag(Gσ(en+1(k)),…,Gσ(en+m(k))) R˜(t−1)(k)=Mr(k)C(y)−1(k)MrT(k) Using Equation (64) to calculate SR˜(t−1)(k) Szz(t−1)(k|k−1)=Tria([Z(k|k−1),SR˜(t−1)(k)]) Pxz(t−1)(k|k−1)=X(k|k−1)ZT(k|k−1) K˜(t−1)(k)=Pxz(t−1)(k|k−1)/Szz(t−1)T(k|k−1)/Szz(t−1)(k|k−1) x^(k|k)t=x^(k|k−1)+K˜(t−1)(k)(z(k)−z^(k|k−1)) if ‖x^(k|k)t−x^(k|k)t−1‖‖x^(k|k)t−1‖≤ε then  x^(k|k)=x^(k|k)t,S(k|k)=Tria([X(k|k−1)−K˜(k)Z(k|k−1),K˜(k)SR˜(k)])  go to step 5. else  *t* = *t* + 1, go to step 4.2 end**5**
*k* = *k* + 1, go to step 3.

In the algorithm, superscript *t* is the fixed-point iteration index and $\hat{x}{\left(k|k\right)}^{t}$ denotes the solution at the fixed-point iteration *t*. A small positive threshold ε is set as the stopping condition, and the fixed-point iterative approach is used to obtain the optimal solution $\hat{x}(k|k)$. Notably, the prior estimate $\hat{x}(k|k-1)$ is set as the initial value for the iterative process, so the algorithm will converge quickly and the computation time will be low [26]. The kernel width σ is a critical parameter, and a large or small kernel width cannot optimize the performance of the algorithm [27]. Considering the accuracy and convergence of the algorithm in this paper, the kernel width was set to σ=5 by manual tuning, and the stopping condition of the fixed-point iterative process was set to $\varepsilon =10^{-6}$.

## 4. Simulation Verification

### 4.1. Simulation Experimental Platform

A co-simulation experimental platform using CarSim and Simulink is designed to verify the validity of the proposed estimation scheme. Figure 5 presents the joint simulation platform used, which consists of four main parts: (a) The vehicle joint simulation system is composed of the CarSim and Simulink modules. CarSim has an internal vehicle dynamics model with 27 degrees of freedom, which can provide inputs and measurements for the state estimator. (b) The vehicle parameter identification system can estimate the vehicle mass and tire forces. (c) The data acquisition system can capture on-board sensor information. (d) The vehicle driving state is finally obtained by the vehicle state estimation system. It is worth mentioning that, because of the high accuracy and high fidelity of the CarSim software, it can provide a sufficiently accurate reference for the estimated vehicle tire force and state [5]. The basic parameters of the vehicle used are shown in Table 1.

In the actual operation of the vehicle, the sensor noise is not always Gaussian noise because the measurement signal is often disturbed by large outliers. In this paper, we assume that both the process and measurement noises of the system are heavy-tailed non-Gaussian noises satisfying a mixed Gaussian distribution with the following covariance matrix.
$$\left\{ \begin{array}{l} Q_{v_{x}}=0.208,v_{k-1,v_{x}}\sim 0.8N(0,0.1^{2})+0.2N(0,1^{2})\\ Q_{v_{y}}=0.208,v_{k-1,v_{y}}\sim 0.8N(0,0.1^{2})+0.2N(0,1^{2})\\ Q_{r}=0.208,v_{k-1,r}\sim 0.8N(0,0.1^{2})+0.2N(0,1^{2}) \end{array}\right.$$
$$\left\{ \begin{array}{l} R_{a_{x}}=0.208,w_{k,a_{x}}\sim 0.8N(0,0.1^{2})+0.2N(0,1^{2})\\ R_{a_{y}}=0.208,w_{k,a_{y}}\sim 0.8N(0,0.1^{2})+0.2N(0,1^{2})\\ R_{r}=0.208,w_{k,r}\sim 0.8N(0,0.1^{2})+0.2N(0,1^{2})\\ R_{v_{c11}}=0.832,w_{k,v_{c11}}\sim 0.8N(0,0.2^{2})+0.2N(0,2^{2})\\ R_{v_{c12}}=0.832,w_{k,v_{c12}}\sim 0.8N(0,0.2^{2})+0.2N(0,2^{2})\\ R_{v_{c21}}=0.832,w_{k,v_{c21}}\sim 0.8N(0,0.2^{2})+0.2N(0,2^{2})\\ R_{v_{c22}}=0.832,w_{k,v_{c22}}\sim 0.8N(0,0.2^{2})+0.2N(0,2^{2}) \end{array}\right.$$
that is
(65)Q=diag([0.208,0.208,0.208])
(66)R=diag([0.208,0.208,0.208,0.832,0.832,0.832,0.832])

### 4.2. Results and Discussion

In this section, two typical scenarios are used to verify the effectiveness of the proposed hierarchical estimation method: 1. double lane change condition; 2. sinusoidal steering condition. Since the vehicle driving state has undergone dramatic changes in these two cases, the above conditions can fully verify the validity of the proposed method. The Dugoff model [8] is a common semi-empirical tire model in vehicle dynamics, and its longitudinal and lateral tire forces are also shown in this paper and compared with the tire forces estimated by ASMO. In addition, conventional Kalman filters, including EKF [11], UKF [18], and CKF [19], are also used to estimate the vehicle driving state for comparison with MCSCKF. It is worth noting that the EKF, UKF, and CKF use the Dugoff tire model, while the MCSCKF uses the ASMO-estimated tire forces. The estimation performance of the different methods is indicated by the root mean square error (RMSE) index.
(67)$$RMSE=\sqrt{\frac{1}{M}\sum_{k=1}^{M}{\left(x(k)-\hat{x}(k)\right)}^{2}}$$
where M is the number of sampling points and k represents the current moment.

#### 4.2.1. Double Lane Change Situation

For this condition, we assume that both process and measurement noise satisfy non-Gaussian distributions, and the corresponding covariance matrices **Q** and **R** are given in Equations (65) and (66). The tire-road friction coefficient is known to be 0.85. The initial state and the corresponding covariance matrix are x(0|0)=[40/3.6,0,0] and P(0|0)=0.01×diag([1,1,1]). Figure 6 illustrates the input and measurement signals of the estimator.

To simulate the case of increased vehicle mass, the initial vehicle mass is set to 1200 kg, while the real mass is 1412 kg. Figure 7 shows the estimation results of the vehicle mass, from which it can be seen that the curve can eventually converge to the real value with small fluctuations, which proves that the proposed FRLS method can accurately identify the vehicle mass under double-lane change conditions. Figure 8 shows the results of the longitudinal and lateral tire forces from the ASMO and Dugoff tire models. As can be seen, the ASMO method can observe the longitudinal and lateral tire forces more accurately than the Dugoff model, especially when the vehicle is turning. This is because when the vehicle is turning, the tire is more likely to be in the nonlinear region, the parameters of the Dugoff model tend to be time-varying, and the calculated tire force error becomes larger. Meanwhile, the Dugoff model has a larger peak error for the estimation of longitudinal tire forces, while the ASMO method is able to follow the true values better. Table 2 shows the RMSE and percentage error of the tire force estimation, from which it can be seen that the ASMO estimates have smaller RMSE and percentage error compared to the Dugoff method. This demonstrates that the proposed ASMO method has higher accuracy and robustness than the Dugoff tire model under double-lane change operation.

It is worth noting that the percentage errors in Table 2 are obtained from the “RMSE/System Maximum Value”. The percentage error indicators that appear below are calculated using the same method.

Figure 9 shows the final vehicle state estimation results for the double-lane change condition. For the estimation of $v_{x}$, the EKF has the largest fluctuation, and the curve has a tendency to diverge, which is most affected by the non-Gaussian noise. The UKF and CKF have similar estimation performances but have large errors from the reference value, especially when the vehicle is turning. This is because when the vehicle turns, the tires tend to show nonlinear characteristics, and the error of the tire force calculated by the Dugoff tire model becomes larger. At the same time, the large outliers in the on-board sensor data increase and the non-Gaussian characteristics of the noise become stronger, and all of the above have a large impact on the estimation results. The proposed MCSCKF shows the best estimation performance with minimum fluctuations and errors under non-Gaussian noise. For the estimation of $v_{y}$ and β, the MCSCKF method still follows the reference value accurately compared to the conventional Kalman filters (EKF, UKF, and CKF). Table 3 shows the RMSE and percentage error of the different methods. For the estimation of the three vehicle states, MCSCKF has the smallest RMSE (0.0015, 0.0092, and 0.0008) and percentage error (0.01%, 4.6%, and 4.5%) compared to the other methods. Table 4 shows fixed-point iterations of the proposed algorithm, and it is clear that MCSCKF has a very small iteration number. The above results demonstrate that the MCC can suppress non-Gaussian noise better than the conventional Kalman filter, and the MCSCKF has higher accuracy and robustness for vehicle state estimation under double-lane change conditions.

#### 4.2.2. Sinusoidal Steering Condition

In this operating condition, the process and measurement noise are assumed to be heavy-tailed non-Gaussian, satisfying a mixed Gaussian distribution. Equations (65) and (66) give the corresponding covariance matrices **Q** and **R**. The tire-road friction coefficient is set to 0.85. To further verify the estimation performance of the proposed method when the vehicle is driving vigorously, the longitudinal speed is increased from 40 km/h to 80 km/h. The initial state vectors and the corresponding covariance matrices are x(0|0)=[80/3.6,0,0] and P(0|0)=0.01∗diag([1,1,1]). Figure 10 illustrates the input and measurement signals of the estimator.

To simulate the vehicle mass decrease, the initial vehicle mass is set to 1600 kg, while the real vehicle mass is 1412 kg. The estimation results for the vehicle mass are given in Figure 11. It can be seen that the curve can quickly converge to near the true value in the first 2 s, and after that, it can follow the reference value with small fluctuations. This proves that the FRLS method can quickly and accurately identify the vehicle’s mass at the start of the vehicle’s operation. The results for the longitudinal and lateral tire forces from the ASMO and Dugoff tire models are presented in Figure 12. As can be observed, the ASMO has smaller errors than the Dugoff model, especially when the vehicle is turning. This is because the Dugoff tire model does not fit the tire nonlinearity well when the vehicle is turning, resulting in larger errors for the calculated tire forces. Table 5 shows the RMSE and percentage error of the tire force estimation, and it is clear that the RMSE and percentage error of the AMSO estimation are smaller compared to the Dugoff method. This also shows that the proposed ASMO method has higher accuracy and robustness than the Dugoff tire model under sinusoidal steering conditions.

Figure 13 shows the vehicle state estimation results under the sinusoidal steering condition. For the three vehicle states, the EKF has the largest estimation error with a tendency to diverge and exhibits the worst accuracy and robustness, while the UKF and CKF have similar performances and do not follow the true values well. The traditional estimation scheme (UKF/CKF + Dugoff) has deteriorated in performance due to the error of the tire model and the interference of non-Gaussian noise. In contrast, the hierarchical estimation method can estimate the vehicle’s driving state more accurately, especially when the lateral motion is significant (lateral acceleration and lateral vehicle speed peaks). The RMSE and percentage error of the different methods are listed in Table 6. It can be seen that the estimation results of MCSCKF have the smallest RMSE (0.0314, 0.1047, 0.0047) and percentage error (0.13%, 12.0%, 11.9%) compared to the other methods under non-Gaussian noise. Table 7 shows the fixed-point iterations of the proposed algorithm, and MCSCKF can quickly obtain the optimal estimate after a few iterations. The above results show that the MCC can better deal with non-Gaussian noise, and the MCSCKF has higher accuracy and robustness for vehicle state estimation under sinusoidal steering conditions.

In summary, the proposed FRLS can accurately identify the vehicle mass when it increases or decreases. Also, the ASMO-based method can observe the longitudinal and lateral tire forces with a small error when the tire is in the nonlinear region. In addition, the MCSCKF can accurately estimate the vehicle’s driving state in non-Gaussian environments. The proposed robust hierarchical estimation scheme has better accuracy and robustness than the conventional estimation methods for vehicle state estimation in practical operating conditions.

## 5. Conclusions

A robust hierarchical estimation scheme for vehicle states was proposed in this paper. First, the vehicle mass was identified by the FRLS and a lateral dynamics model. Secondly, the longitudinal and lateral tire forces were accurately observed by ASMO based on the wheel rotation dynamics models and the single-track model. Finally, using the estimated information, the vehicle’s driving state containing the longitudinal vehicle speed, the lateral vehicle speed, and the sideslip angle were estimated by MCSCKF in non-Gaussian scenarios. To validate the effectiveness of the proposed hierarchical estimation scheme, two typical operating conditions were tested based on a joint simulation platform. The results showed that the proposed FRLS and ASMO perform well in the estimation of mass and tire forces. Also, the MCSCKF can accurately estimate the vehicle’s driving state under non-Gaussian conditions. The proposed robust hierarchical estimation scheme can cope better with model parameter ingestion and non-Gaussian noise than conventional estimation methods and has great application potential in the field of autonomous driving perception. However, the proposed method is not applied to the real vehicle test; the complexity and real-time performance of the algorithm need to be focused on, and the dynamics model does not consider the rolling motion of the vehicle. The problems mentioned above will be the main direction of our future research.

## Figures and Tables

**Figure 1 entropy-25-00453-f001:**
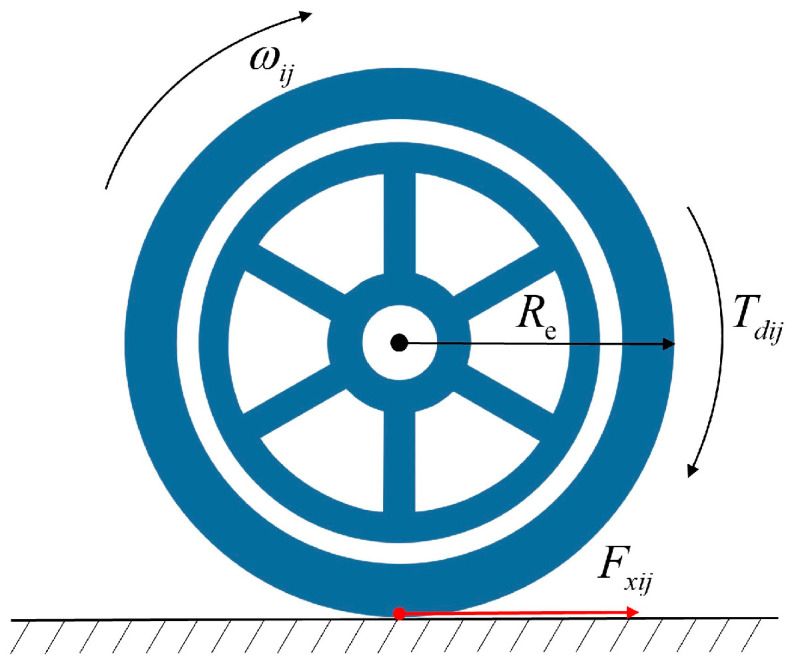
Single-wheel rolling dynamics model.

**Figure 2 entropy-25-00453-f002:**
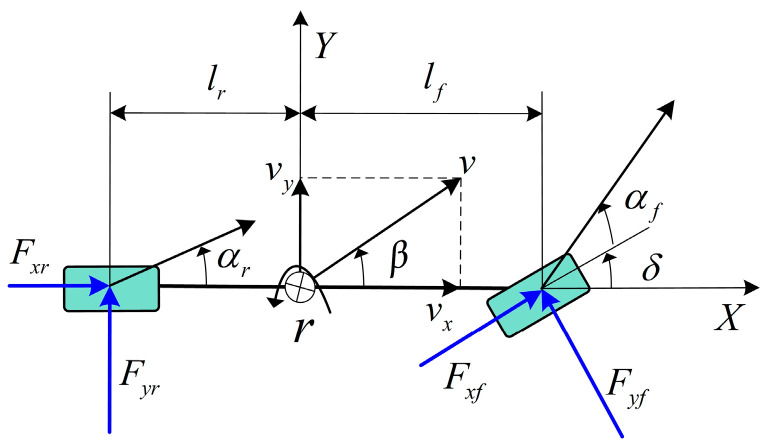
Single-track plane model.

**Figure 3 entropy-25-00453-f003:**
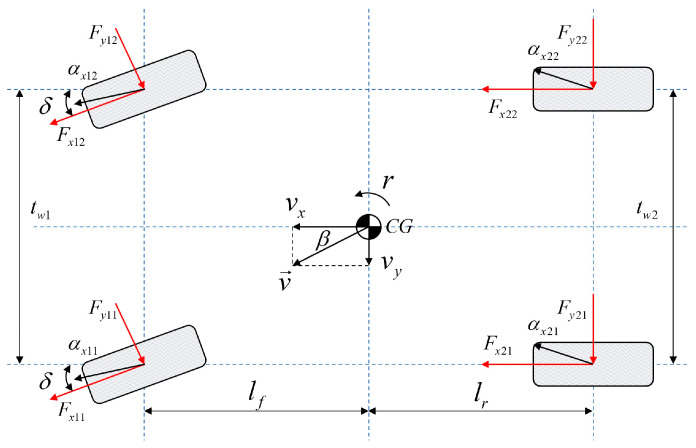
Schematic diagram of the four-wheeled vehicle model.

**Figure 4 entropy-25-00453-f004:**
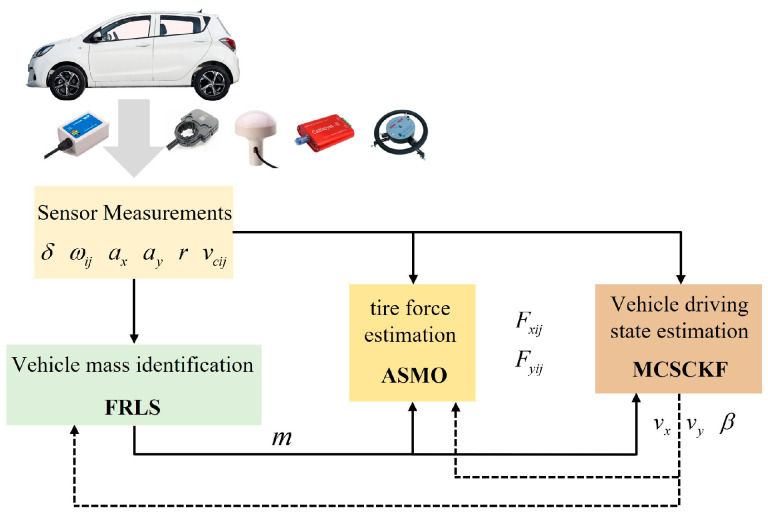
A robust hierarchical estimation scheme for vehicle state.

**Figure 5 entropy-25-00453-f005:**
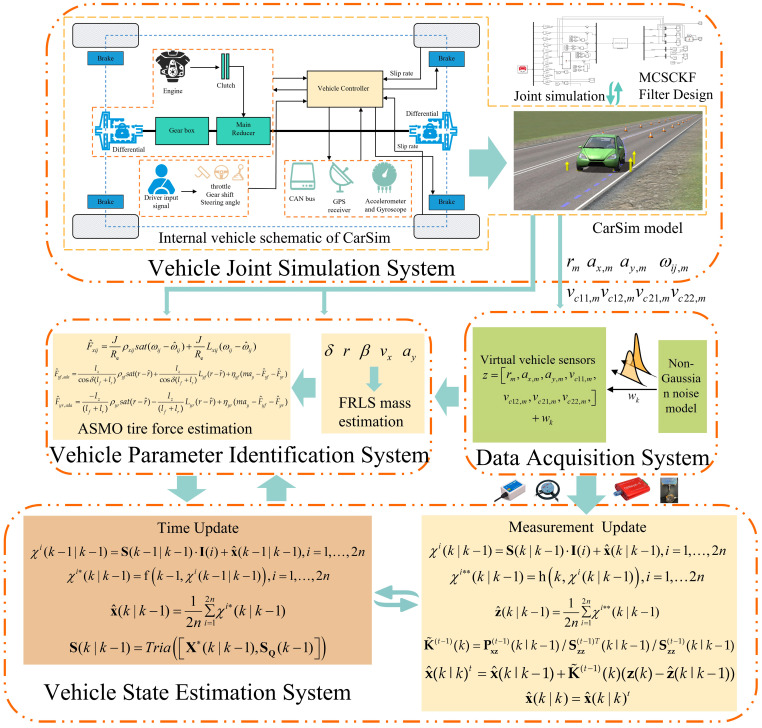
Simulation experimental platform of a robust hierarchical estimation scheme for vehicle state.

**Figure 6 entropy-25-00453-f006:**
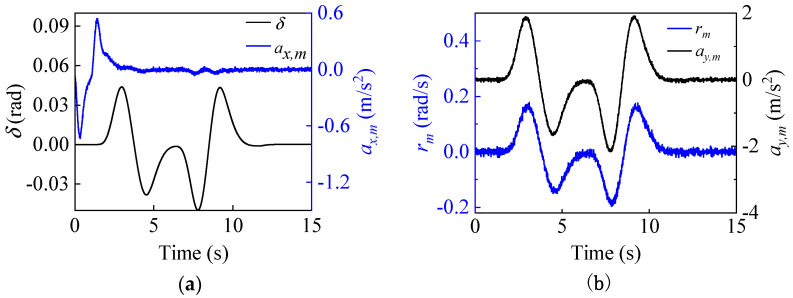
The input and measurement signals of the estimator. Input: (**a**) Front wheel angle δ, longitudinal acceleration $a_{x,m}$. Measurement: (**b**) yaw rate $r_{m}$ and lateral acceleration $a_{y,m}$ under double-lane change conditions.

**Figure 7 entropy-25-00453-f007:**
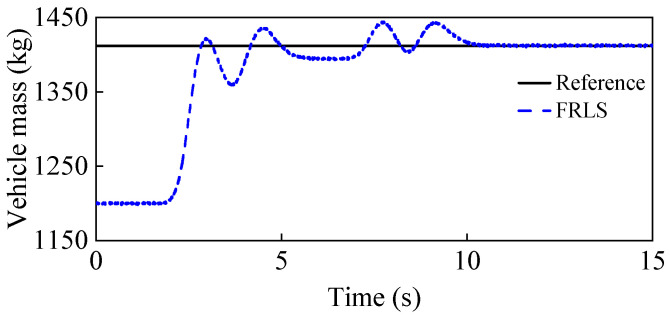
Vehicle mass estimation results under a double-lane change condition.

**Figure 8 entropy-25-00453-f008:**
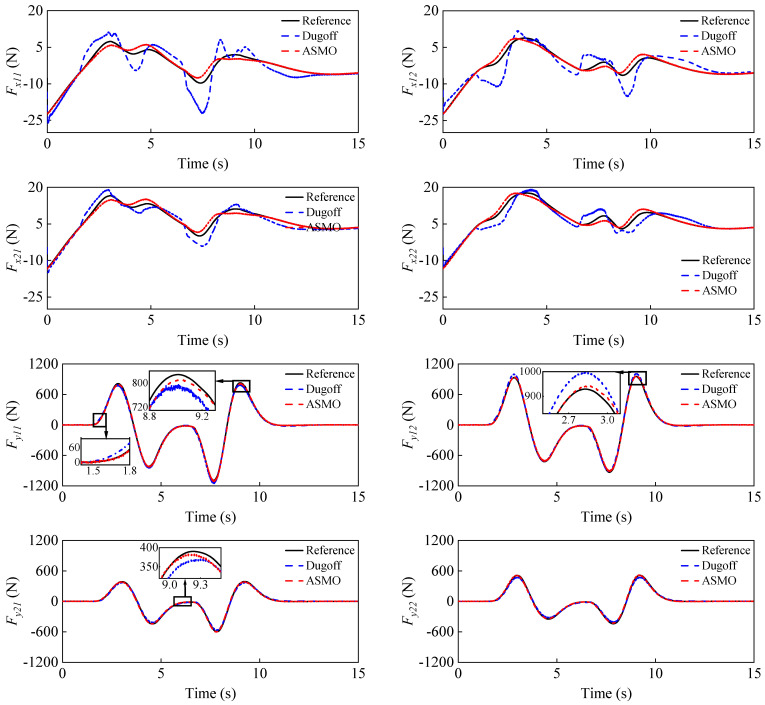
Longitudinal and lateral tire force estimation results under a double-lane change condition.

**Figure 9 entropy-25-00453-f009:**
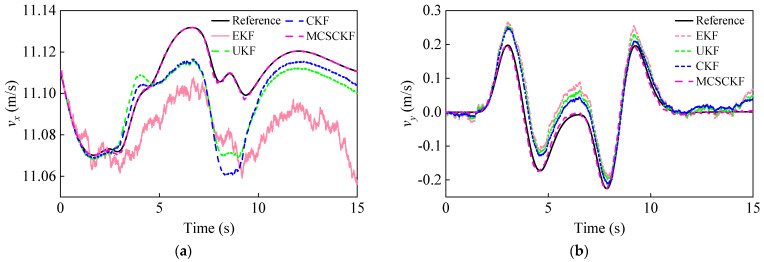
Vehicle state estimation results under a double-lane change condition. (**a**) Longitudinal vehicle velocity estimation results; (**b**) Lateral vehicle velocity estimation results; (**c**) Sideslip angle estimation results.

**Figure 10 entropy-25-00453-f010:**
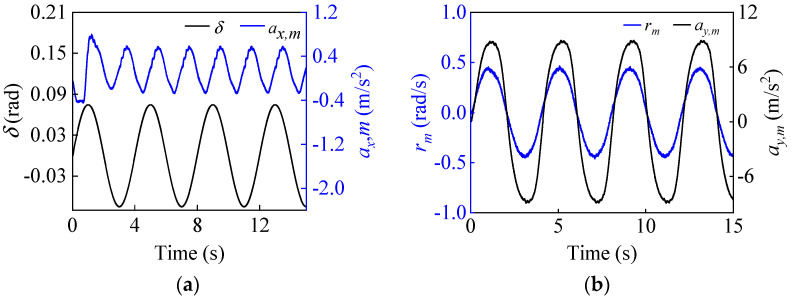
The input and measurement signals of the estimator. Input: (**a**) Front wheel angle δ, longitudinal acceleration $a_{x,m}$. Measurement: (**b**) yaw rate $r_{m}$ and lateral acceleration $a_{y,m}$ under sinusoidal steering conditions.

**Figure 11 entropy-25-00453-f011:**
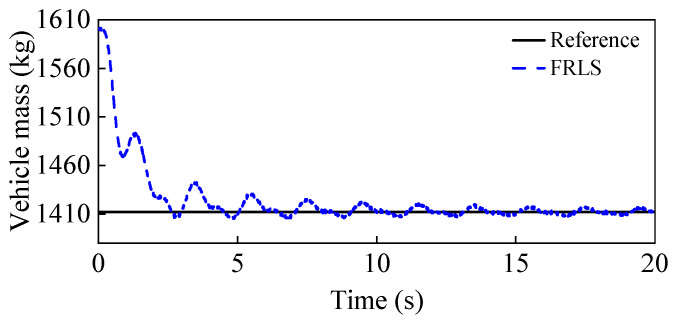
Vehicle mass estimation results under a sinusoidal steering condition.

**Figure 12 entropy-25-00453-f012:**
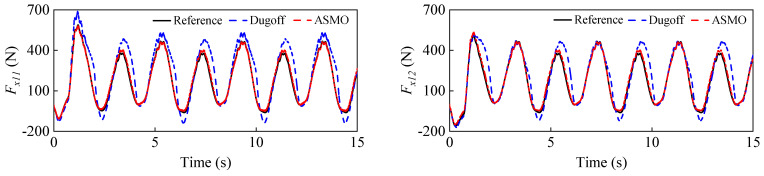
Longitudinal and lateral tire force estimation results under a sinusoidal steering condition.

**Figure 13 entropy-25-00453-f013:**
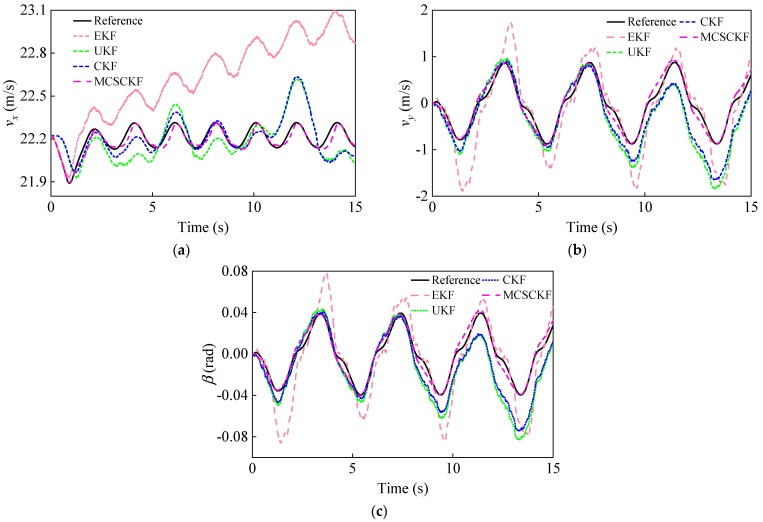
Vehicle state estimation results under a sinusoidal steering condition. (**a**) Longitudinal vehicle velocity estimation results; (**b**) Lateral vehicle velocity estimation results; (**c**) Sideslip angle estimation results.

**Table 1 entropy-25-00453-t001:** Basic parameters of the vehicle.

Vehicle Parameter	Value	Unit
Sprung mass ($m_{s}$)	1270	kg
Unsprung mass ($m_{us}$)	142	kg
Vehicle mass (m)	1412	kg
Yaw moment of inertia ($I_{zz}$)	1536.7	kg·m^2^
Distance from the front and rear axles to the CG ($l_{f}/l_{r}$)	1.015/1.895	m
The wheelbase of the front and rear wheels ($t_{w1}/t_{w2}$)	1.675/1.675	m
The effective rolling radius of the tire ($R_{e}$)	0.325	m
Height of CG ($h_{g}$)	0.54	m
**Sensor Parameter**		
**Type of Sensor**	**Standard Deviation (** σ **)**	
	**Value**	**Unit**
INS (Inertial Navigation System) sensor (100 Hz)		
Yaw rate gyroscope	$1\times 10^{-1}$	deg/s
Longitudinal accelerometer	$1\times 10^{-1}$	m/s^2^
Lateral accelerometer	$1\times 10^{-1}$	m/s^2^
CAN bus (50 Hz)		
Steering wheel sensor	$1\times 10^{-1}$	deg
Wheel speed sensor	$2\times 10^{-1}$	m/s

**Table 2 entropy-25-00453-t002:** RMSE and percentage error of the longitudinal and lateral tire forces estimation under double-lane change conditions.

Method	Tire Force	*F_x_* _11_	*F_x_* _12_	*F_x_* _21_	*F_x_* _22_	*F_y_* _11_	*F_y_* _12_	*F_y_* _21_	*F_y_* _22_
System Maximum Values (N)		7.3	8.7	16.5	17.6	828.1	956.3	390.1	512.4
Dugoff	RMSE (N)	3.34	3.10	1.57	1.77	17.30	19.17	11.35	14.61
	Percentage Errors (%)	45.8	35.6	9.5	10.1	2.1	2.0	2.9	2.9
ASMO	RMSE	0.98	0.92	0.86	0.81	10.75	11.42	8.90	9.23
	Percentage Errors (%)	13.4	10.6	5.2	4.6	1.3	1.2	2.3	1.8

**Table 3 entropy-25-00453-t003:** RMSE and percentage error of different methods under double-lane change conditions.

Filter		$v_{x}$ (m/s)	$v_{y}$ (m/s)	β (rad)
	System Maximum Values	11.13	0.20	0.0179
EKF	RMSE	0.0353	0.0447	0.0040
	Percentage Errors (%)	0.32	22.4	22.3
UKF	RMSE	0.0140	0.0345	0.0031
	Percentage Errors (%)	0.13	17.2	17.3
CKF	RMSE	0.0142	0.0277	0.0025
	Percentage Errors (%)	0.13	13.9	14.0
MCSCKF	RMSE	0.0015	0.0092	0.0008
	Percentage Errors (%)	0.01	4.6	4.5

**Table 4 entropy-25-00453-t004:** Fixed-point iterations of the MCSCKF algorithm for double-lane change conditions.

Filter	Average Iteration Number	Maximum Iteration Number	Minimum Iteration Number
MCSCKF	2.0218	3	1

**Table 5 entropy-25-00453-t005:** RMSE and percentage error of the longitudinal and lateral tire forces estimation under sinusoidal steering conditions.

Method	Tire Force	*F_x_* _11_	*F_x_* _12_	*F_x_* _21_	*F_x_* _22_	*F_y_* _11_	*F_y_* _12_	*F_y_* _21_	*F_y_* _22_
System Maximum Values (N)		587.6	511.8	358.4	358.3	2303.0	5730.5	801.0	4162.6
Dugoff	RMSE (N)	90.41	79.87	54.11	58.67	686.73	728.52	431.74	420.31
	Percentage Errors (%)	15.4	15.6	15.1	16.4	29.9	12.7	54.0	10.1
ASMO	RMSE	13.40	15.04	12.07	12.83	312.95	317.89	194.06	172.42
	Percentage Errors (%)	2.3	2.9	3.4	3.6	13.6	5.5	24.2	4.1

**Table 6 entropy-25-00453-t006:** RMSE and percentage error of different methods under a sinusoidal steering condition.

Filter		$v_{x}$ (m/s)	$v_{y}$ (m/s)	β (rad)
	System Maximum Values	23.31	0.87	0.0394
EKF	RMSE	0.4931	0.4781	0.0216
	Percentage Errors (%)	2.1	55.0	54.8
UKF	RMSE	0.1372	0.4428	0.0199
	Percentage Errors (%)	0.59	51.0	50.5
CKF	RMSE	0.1202	0.3685	0.0166
	Percentage Errors (%)	0.52	42.4	42.1
MCSCKF	RMSE	0.0314	0.1047	0.0047
	Percentage Errors (%)	0.13	12.0	11.9

**Table 7 entropy-25-00453-t007:** Fixed-point iterations of the MCSCKF algorithm for sinusoidal steering conditions.

Filter	Average Iteration Number	Maximum Iteration Number	Minimum Iteration Number
MCSCKF	2.2115	4	1

## Data Availability

The data that support the findings of this study are available from the corresponding author, Yongbin Li, upon reasonable request.

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
