# Peer review of "A Robust Hierarchical Estimation Scheme for Vehicle State Based on Maximum Correntropy Square-Root Cubature Kalman Filter"

_entropy, 2023, doi:10.3390/e25030453_

Round 1
Reviewer 1 Report
The writing of this paper is good, but the contents have a major weakness: no experimental validation.
1、This paper is only verified by simulation simulation, and lacks actual experiments to verify the accuracy of the model.
2、Simulation model parameters should be set more precisely, such as considering the rolling motion of the vehicle.
3、The conclusion of the simulation part is relatively simple.
4、The complexity of the algorithm is not enough.
5、The conclusion can only be drawn from the dynamics model, which is lack of guidance for practical application cases.
Reviewer 2 Report
The paper designs an estimation scheme for vehicle velocity and sideslip angle. On-board sensors are used and auxiliary estimates of vehicle mass and tire forces are constructed. The manuscript needs clarification in the theoretical part as well as in the experimental part. Although the contribution is publishable, the topic does not match the scope of the Entropy journal. It should be rejected from this journal and forwarded to another one.
The introduction provides adequate background. The vehicle models are clearly presented. The argument that the vehicle mass can vary is valid, since the number of passengers can vary and various load can be carried. However, this could also shift the centre of gravity considerably and change the moment of inertia. The need to estimate the mass and the possibility to neglect the related parameters should be supported by experiments or citations to literature.
Section 3.2 lacks citations. The description is not clear. In line 231, it is not clear, what is constructed (Eq. (13) cannot be constructed in a different form; based on the equation, an observer can be constructed). The psi_s disturbance term seems to be unknown and therefore, it cannot be used in (16), (23) and in the related equations. In (23) and other equations (including Figure 5), there is the S term and also the state x_s. These terms have to be replaced by variables that are available to the observer, i.e. y_s and hat{x}_s. Note also that the S term appears in many other equations, but with the meaning of omega-hat{omega} or r-hat{r}. Further, there are conflicting equations (23)/(25) and (30)/(31) that are not distinguished well. Why do they differ? Which ones are correct? The terms hat{omega}_ij in (28) and hat{r} in (30) have to be defined in the paper.
Section 3.3 introduces correntropy of two random variables in (39). For multiple independent samples governed by the corresponding distributions, an estimate is designed in (41). The correntropy criterion in (57) seems to be a misnomer, since it considers multiple different random variables with possibly different distributions. Although the name can be inappropriate, the authors can choose any criterion at will.
The term "cost function" used in lines 329, 331, 368 is incorrect, since it is maximised. Utility or reward would be better. A generic term "objective function" is also appropriate. The notation S_xz in (51) is a little bit confusing, since it appears to denote a "square root" of the cross-covariance. In (62), it is not clear which variable is the "updated covariance matrix" related to. The reason of this "update" is not clear as well.
The assumption x=hatx made in line 381 makes no sense. The motivation of using such a simplification has to be explained. Equations (63), (64) use undefined symbols and do not seem correct. Is S*Cx^{-1}*S^T meant in (63)? Where is (63) used? Is M_R from (64) used to construct S_R in (65) directly? The whole part between lines 381 and 385 has to be clarified. The while cycle in the Algorithm is unclear, since no term seems to be updated. Where is hatx(k|k)^t used there? Can S_xz be computed outside the loop?
The presentation of the results should be improved. What is meant by "percentage error"? Why are "maximum values" important and what do they represent? Do they relate the system or the estimators? (See that they are the same for all filters.) In lines 459-461, the part of the sentence after "while" is not clear. In Figures 8 and 12, same axes limits should be used for an easier comparability. In Figure 12, the reason of having a noisy reference trajectory should be explained. Also, it is not clear whether the sinusoidal steering is realistic or not. How do the position and velocity trajectories look like?
There are several typos.
*(12) is a copy of (11) - covariance update is missing
*line 228: define ... is -> define ... as
*lines 238, 252: a symbolic function -> the signum function (sign function)
*line 297: measurement
*line 319: measure
*(59) - proportionality holds (in the place of the first =)
*(62) - C(k)^{-1}
*caption of Figure 10: sinusoidal steering
Reviewer 3 Report
This paper presents a robust hierarchical estimation technique for vehicle states, particularly vehicle mass, longitudinal and lateral tire forces, longitudinal and lateral speeds, and the sideslip angle.
Its effectiveness has been tested by resorting to a simulation platform based on CarSim and Mathworks Simulink.
The paper is scientifically sound.
I do not feel fully qualified to evaluate the proposed mathematical model, but the presentation seems complete and well-structured.
Round 2
Reviewer 1 Report
The authors have addressed many of the previous comments.
Reviewer 2 Report
The authors have addressed many of the previous comments, but there remains several points to fix.
The construction of estimates has to be explicit. There have to be equations that define hat{d}, hat{r}, hat{omega}_ij, hat{F}_yf and hat{F}_yr; it does not suffice to say that hat{omega}_ij is an estimate of omega_ij and that hat{r} is an estimate of r (see lines 269 and 276). There has to be a single equation that defines hat{d}; see that (23) and (25) make different claims. Also, hat{F}_yf and hat{F}_yr have to be defined exactly once; see that (30) and (31) differ in the presence of the last addend. If the authors wish to use both equations, there have to be different symbols on the left hand sides and these symbols have to be explained.
The symbol S can still be found in equations (23), (24), (25). It should be replaced by y_s-hat{x}_s there. Next, the symbol S can still be found in equation (28), where it has to be replaced by omega_ij-hat{omega}_ij; it does not have the meaning x_s-hat{x}_s there. Similarly, the symbol S in equations (30) and (31) has to be replaced by r-hat{r}; it does not have the meaning x_s-hat{x}_s there. Finally, the symbols S in the "Vehicle Parameter Identification System" block in Figure 5 have to be replaced by omega_ij-hat{omega}_ij or by r-hat{r}. Remind also that hat{omega}_ij and hat{r} have not been defined so far and that they have to be based on some measured quantities.
All estimates have to be constructed from quantities that are available. Although an interference terms psi can be expressed as a multiple of F_xf, it cannot be used to construct the estimate (16), since the force F_xf is nowhere measured. The relation psi=k*F_xf can be used in a theoretical analysis, but it cannot be used to construct a function in a physical device that processes real-world signals. If the term psi is available, there have to be an equation that shows how it is constructed from available measurements and estimates. If the term psi is not available in (16) and (23), the analysis in (16), (17) and (24) could be done for the assumption psi=0 and could be followed by a short discussion.
The relation x(k)=hat{x}(k|k-1) claimed on line 392 is not valid, if the symbol x(k) denotes the true state, which is unknown. The estimate hat{x} cannot be constructed as being equal to the true state x(k) and vice versa, the true state x(k) cannot be expected to be equal to an estimate. A candidate for the argument of the optimum in the objective function (57) is searched; according to point 4.1 of the Algorithm:MCSCKF, the construction hat{x}(k|k)^(0)=hat{x}(k|k-1) is used. The equations are evaluated with respect to the candidates; they are not evaluated with respect to the unknown true state of the system.
According to (60) and (62), the first line of (63) should contain M*Cx^{-1}*M^T instead of M*I*M^T. Nevertheless, the term tilde(P)(k|k-1) is nowhere used; thus the line can be removed. Since C_(y) is a diagonal matrix according to (60), the square root in (64) is given according to (63) as M_r(k)*{C_(y)(k)}^{-1/2}, where the inverse of the square root can be applied directly on the diagonal elements of C_(y)(k). In Algorithm:MCSCKF, the term M_r(k) computed in point 4.2 should be computed already in point 4.1, since it is the same for all iterations.
Since the maximum values are related to the system, they should occupy their own line for "System", while the "Methods" in Tables 2 and 5 and "Filters" in Tables 3 and 6 should not repeat the numbers. The definition of "percentage errors" as "RMSE/maximal_value" has to be introduced in the manuscript. Nevertheless, this evaluation metric is dubious, since "RMSE" is taken as an average over a time interval, whereas the maximal values are related to single time instants. The vertical axes limits in Figures 8 and 12 should be chosen to be the same for the left wheel and the corresponding right wheel; e.g. the same for F_x11 and F_x12, for F_x21 and F_x22 etc. It would be even better to have the same axes limits for all longitudinal and all lateral forces, i.e. same axes for F_x11, F_x12, F_x21 and F_x22 and same axes for F_y11, F_y12, F_y21 and F_y22. This would enable the reader to have a direct graphical comparison.
